# High Affinity of Nanoparticles and Matrices Based on Acid-Base Interaction for Nanoparticle-Filled Membrane

Tsutomu Makino [1], Keisuke Tabata [1], Takaaki Saito [1], Yosimasa Matsuo [1] and Akito Masuhara [1,2,*]

[1] Graduate School of Science and Engineering, Yamagata University, 4-3-16, Jonan, Yonezawa 992-8510, Yamagata, Japan; t221366m@st.yamagata-u.ac.jp (T.M.); t211992d@st.yamagata-u.ac.jp (K.T.); t221575m@st.yamagata-u.ac.jp (T.S.); t231392m@st.yamagata-u.ac.jp (Y.M.)

[2] Frontier Center for Organic Materials (FROM), Yamagata University, 4-3-16, Jonan, Yonezawa 992-8510, Yamagata, Japan

* Correspondence: masuhara@yz.yamagata-u.ac.jp; Tel.: +81-238-26-3891

**Abstract:** The introduction of nanoparticles into the polymer matrix is a useful technique for creating highly functional composite membranes. Our research focuses on the development of nanoparticle-filled proton exchange membranes (PEMs). PEMs play a crucial role in efficiently controlling the electrical energy conversion process by facilitating the movement of specific ions. This is achieved by creating functionalized nanoparticles with polymer coatings on their surfaces, which are then combined with resins to create proton-conducting membranes. In this study, we prepared PEMs by coating the surfaces of silica nanoparticles with acidic polymers and integrating them into a basic matrix. This process resulted in the formation of a direct bond between the nanoparticles and the matrix, leading to composite membranes with a high dispersion and densely packed nanoparticles. This fabrication technique significantly improved mechanical strength and retention stability, resulting in high-performance membranes. Moreover, the proton conductivity of these membranes showed a remarkable enhancement of more than two orders of magnitude compared to the pristine basic matrix, reaching $4.2 \times 10^{-4}$ S/cm at 80 °C and 95% relative humidity.

**Keywords:** RAFT PwP; core–shell nanoparticles; MMMs; acid-base interaction; proton exchange membranes

## 1. Introduction

Mixed matrix membranes (MMMs) [1,2] are composite structures composed of nanoparticles and a polymer matrix, which exhibit features of both materials. Consequently, MMMs find applications in various fields, including gas separation [3,4], antimicrobial [5,6], and proton exchange membranes (PEMs) [7,8]. In particular, PEMs selectively conduct protons from anode to cathode for converting chemical energy to electrical energy, and this is employed in proton exchange membrane fuel cells (PEMFCs) [9–12]. PEMFCs are one of the most attractive energy generation systems due to their cleanliness as it uses only hydrogen as fuel, high energy conversion efficiency, low-temperature operation, and compact cell design. In several decades, the functionalization of PEMs has been reported by incorporating nanoparticles. Metal-organic frameworks (MOFs) have been introduced to sulfonated poly (ether ketone) to establish continuous proton-conductive channels through the pores, resulting in improved proton conductivity [7]. Furthermore, sulfonated titanium dioxide nanoparticles have been added to the sulfonated polyethersulfone to improve water absorption and proton conductivity [8]. As evidenced by studies of microphase-separated structures, it has been attributed to the construction of proton-conductive channels that allows for fast proton conduction by introducing nanoparticles [13]. As mentioned earlier, the homogeneous dispersion of nanomaterials in the matrix enhances various functionalities compared to the neat matrix. However, the dispersibility of nanoparticles tends to

decrease with increasing particle concentration due to the aggregation of nanomaterials, resulting in the meandering or blocking of ion-conductive channels [14]. This causes a decrease in proton conductivity and other physical properties.

Previously, we prepared core–shell nanoparticles coated with a block copolymer in order to efficiently disperse them in thermoplastic resins as a matrix [15,16]. The block copolymer was composed of acidic polymer with an affinity for core nanoparticles and hydrophobic polymer that ensures compatibility with the matrix. Specifically, we coated a block copolymer of poly(vinylphosphonic acid)-*b*-polystyrene (PVPA-*b*-PS) for the surface of a cellulose nanocrystal by using a unique polymer coating method named RAFT PwP [17,18]. The obtained core–shell nanomaterials were mixed with polycarbonate to fabricate a free-standing membrane. Compared to only PVPA-coated cellulose nanocrystals, the block copolymer-coated cellulose nanocrystals successfully suppressed aggregation, owing to the affinity of PS and polycarbonate. In addition, the proton conductivity of the membrane was $1.8 \times 10^{-2}$ S/cm at 60 °C and at 95% relative humidity (RH) by the contribution of PVPA.

In this study, we fabricated a nanoparticle-filled membrane that not only integrates the dispersibility of nanoparticles and proton conductivity but also incorporates mechanical strength and retention stability. We have employed acid-base interaction between the surface of nanoparticles and matrices, to achieve well dispersibility and retention stability of nanoparticles (Scheme 1). Specifically, poly(styrenesulfonic acid) (PSSA) was coated onto the surface of $SiO_2$ (silica) nanoparticles by using RAFT PwP (silica@PSSA); also, the basic matrix consists of poly(1-vinyl imidazole)-*co*-poly(butyl acrylate) (P1VIm-*co*-PBA) which was polymerized via free radical polymerization. P1VIm indicated basicity and acted as a proton acceptor from acidic PSSA, resulting in the hydrogen bond being formed. The acid-base interaction based on proton transfer from the PSSA to P1VIm provided an affinity between the silica@PSSA and the P1Vim-*co*-PBA, and the immobilization of silica@PSSA in the matrix that enabled the formation of a free-standing membrane.

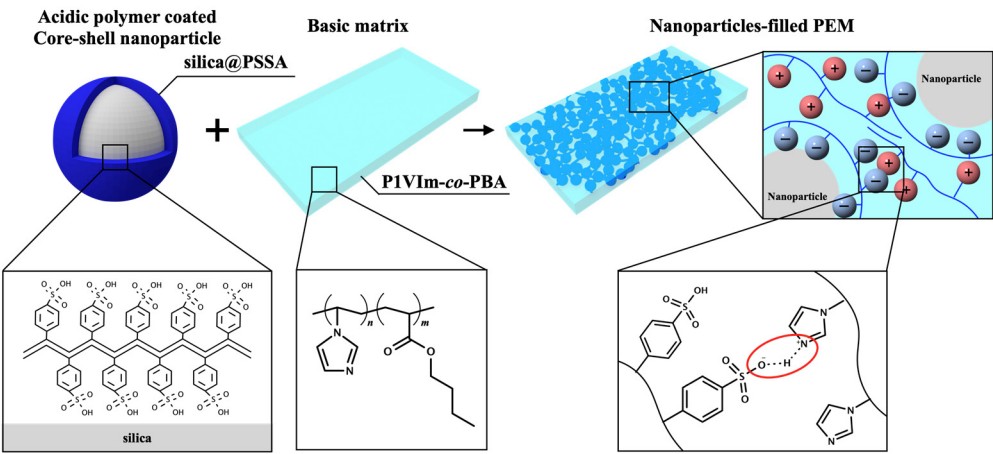

**Scheme 1.** Schematic image of PEM using acid-base interaction.

## 2. Experimental Section

### 2.1. Materials

*N*,*N*-dimethylformamide (DMF, >99.5%), 1-vinylimidazole (1VIm, >98.0%) and butyl acrylate (BA, >99.0%) were purchased from Tokyo Chemical Industry Co., Ltd (Tokyo, Japan). p-styrenesulfonic acid sodium salt (SSNa), 2,2′-azobis (isobutylnitrile) (AIBN, >98.0%) and 4-cyano-4-[(dodecylsulfanylthiocarbonyl) sulfanyl] pentanoic acid (CDSSP, >97.0%) were purchased from FUJIFILM Wako Pure Chemical Corporation (Osaka, Japan). Pristine silica nanoparticles (SEAHOSTAR KE-P10, 100 nm in diameter) were purchased from Nippon Shokubai Co., Ltd. (Osaka, Japan). Cation exchange resin (DIAION$^{TM}$, SK1BH) was purchased from Mitsubishi Chemical Group Corporation (Tokyo, Japan) (Figure S1).

## 2.2. Synthesis of Silica@PSSNa and Silica@PSSA

Pristine silica (0.500 g), SSNa (0.300 g, 1.54 mmol), AIBN (2.50 mg, $1.52 \times 10^{-2}$ mmol), CDSSP (17.5 mg, $4.34 \times 10^{-2}$ mmol), and DMF (5.00 g, 5.25 mL) were added into the test tube and irradiated with ultrasonic waves to disperse. Subsequently, silica@PSSNa was synthesized at 80 °C with stirring for 24 h in the glovebox by the RAFT PwP method. After the polymerization, obtained silica@PSSNa which PSSNa is coated on the silica surface were dispersed in methanol which removed monomers and polymers that remained in the DMF. This process was carried out three times and the samples were dried overnight in a vacuum oven. Silica@PSSNa (0.300 g) and DIAION (1.50 g) were added to the methanol and cation exchange was performed by stirring at room temperature for 2 h. Then, DIAION and nanoparticles were separated by a sieve with a mesh size of 160 μm and silica@PSSA was obtained.

## 2.3. Preparation of P1VIm-co-PBA

1VIm (1.20 g, $1.28 \times 10^{-2}$ mol) and BA (1.10 g, $8.58 \times 10^{-3}$ mol) were copolymerized in DMF (7.00 g, 7.35 mL) by free radical polymerization to generate a copolymer P1VIm-*co*-PBA. After the polymerization, obtained P1VIm-*co*-PBA were dispersed in water and acetone to remove monomers that remained in the DMF.

## 2.4. Fabrication of Silica@PSSNa/P1VIm-co-PBA and Silica@PSSA/P1VIm-co-PBA

Composite membrane was fabricated by the simple method of compounding core–shell nanoparticles and P1VIm-*co*-PBA and casting them in a Teflon Petri dish. Silica@PSSA (0.200 g) was dispersed in methanol and P1VIm-*co*-PBA (0.300 g) was dissolved in methanol (3.00 mL). The dispersion and solution were mixed with stirring for 30 min at room temperature to be a 40 wt% of particle concentration. Finally, the resulting dispersion was cast on a Teflon Petri dish and dried at 60 °C for 3 h and silica@PSSNa/P1VIm-*co*-PBA and silica@PSSA/P1VIm-*co*-PBA was obtained.

## 2.5. Characterization

$^1$H nuclear magnetic resonance ($^1$H NMR; JEOL, Tokyo, Japan) was used to determine the progress of copolymerization and calculation of the molar ratio of P1VIm-*co*-PBA. Ultraviolet-visible spectroscopy (UV-vis JASCO V-670) was used to identify the CTA, which was coated onto the surface of silica nanoparticles. Scanning electron microscopy (SEM, JSMIT800, JEOL) and SEM-energy dispersive X-ray spectrometry (SEM-EDX, SU-8000, Hitachi High-Tech Corp., Tokyo, Japan) measurements were performed to observe the surface morphology. Fourier transform infrared spectroscopy (FT-IR, FT/IR-4700, JASCO, Tokyo, Japan) was measured by the KBr method for nanoparticles and the ATR method for membranes.

## 2.6. Retention Stability of Nanoparticles

The retention stability of the nanoparticles was calculated from the weight difference of the composite membrane after immersion in distilled water for 5 days by using Formula (1), where $W$ (%) is the weight of the membrane. $W_{dry1}$ (g) is the weight of its original membranes dried. $W_{dry2}$ (g) is the weight of the membrane after immersing it in distilled water and drying it.

$$W = \frac{W_{dry2}}{W_{dry1}} \times 100 \tag{1}$$

## 2.7. Proton Conductivity

The proton conductivities were measured for pellet state and membrane state samples under different relative humidities (RH: 65–95% RH) at 80 °C and 85% RH at different temperatures (40–80 °C) in an environment control machine (SH-241, ESPEC, Osaka, Japan) using ac impedance measurements (IM 3570, HIOKI Corp, Nagano, Japan); frequency, $4.6$–$4.6 \times 10^6$ Hz; four terminal method). The proton conductivity was calculated using

Formula (2), where σ is the proton conductivity (S/cm), $R_s$ (Ω) is the resistance obtained from the Cole–Cole plots, $d$ (cm) is the electrode distance, and $S$ (cm$^2$) is the sectional area of sample.

$$\sigma = \frac{1}{R_s} \times \frac{d}{s} \tag{2}$$

## 3. Results and Discussion

### 3.1. Characterization of Core-Shell Nanoparticles, Basic Matrix, and Composite Membrane

SEM-EDX mappings targeting silicon (Si), sulfur (S), and sodium (Na) of silica, silica@PSSNa, and silica@PSSA were shown in Figure 1a. Silica@PSSA exhibited mappings of S derived from PSSA at the same position as the Si derived from silica nanoparticles and also did not show the mapping of sodium (Na) originated from PSSNa, meaning that the cation exchange at the surface of the nanoparticle was successfully progressed. Additionally, the UV-vis absorption of the RAFT agent (CDSSP) from the core–shell nanoparticles was evident in the progression of RAFT PwP (Figure 1b). The polymerization of P1VIm-*co*-PBA was determined by $^1$H NMR (Figure 1c). The copolymer exhibited a broad signal of the main chain at 0.96–2.2 ppm, heterocyclic ring at 6.7–7.7 ppm, and the methylene protons and methyl protons of BA at 3.5–4.1 ppm and 0.60–0.89 ppm [19,20]. Moreover, based on the integral calculations of the peak areas of a, b, c, and d, the molar content of the imidazole group was found to be 40%. From the FTIR spectra of each nanoparticle, peaks of Si-O-Si and Si-O attributed to silica nanoparticles were observed at 1000 and 800 cm$^{-1}$ [21,22], (Figure 1d). The broad absorption of PSSA was observed at 1046 and 830 cm$^{-1}$ [23–25], stretching vibrations of the imidazole rings were assigned to 3114 cm$^{-1}$ [26,27], and C=O and C-O-C stretching vibrations of BA were observed in 1732 and 1111 cm$^{-1}$ [28]. Furthermore, the protonated imidazole (ImH$^+$) peak was observed at approximately 1580 cm$^{-1}$ [29] from only silica@PSSA/P1VIm-*co*-PBA and it was evidence of the proton transfer from the PSSA to P1VIm on the surface of the silica nanoparticles.

### 3.2. Effects of Acid-Base Interaction

To investigate the dispersibility of nanoparticles, photographs and SEM images of silica@PSSNa/P1VIm-*co*-PBA and silica@PSSA/P1VIm-*co*-PBA are shown in Figure 2. According to the SEM image of silica@PSSNa/P1VIm-*co*-PBA, aggregations and voids were clearly observed from the surface of the membrane. This is attributed to the inability of silica@PSSNa/P1VIm-*co*-PBA to form acid-base interactions between the nanoparticles and the matrix. In contrast, silica@PSSA/P1VIm-*co*-PBA showed a smooth surface in which nanoparticles are uniformly dispersed with no aggregations and voids. It was due to the acid-base interaction that enabled the nanoparticles to disperse throughout the matrix rather than aggregate with each other. Furthermore, the membrane densities were calculated from the weights and volumes of the membrane; silica@PSSNa/P1VIm-*co*-PBA was 0.871 g/cm$^3$ and silica@PSSA/P1VIm-*co*-PBA was 1.06 g/cm$^3$ (Table 1). The cross-section SEM image of silica@PSSA/P1VIm-*co*-PBA also confirmed the high dispersion of nanoparticles, as did the surface SEM images (Figure S2). Based on the density of the membrane, the silica@PSSA/P1VIm-*co*-PBA achieved a higher density packing of core–shell nanoparticles.

**Table 1.** Summary of membrane properties of silica@PSSA/P1VIm-*co*-PBA and silica@PSSNa/P1VIm-*co*-PBA.

|  | Density [g/cm$^3$] | Stress [MPa] | Strain [%] | Weight of Membrane [%] |
|---|---|---|---|---|
| silica@PSSA/P1VIm-*co*-PBA | 1.06 | 17.0 | 73.5 | 97.1 |
| silica@PSSNa/P1VIm-*co*-PBA | 0.871 | 7.26 | 58.6 | 71.1 |

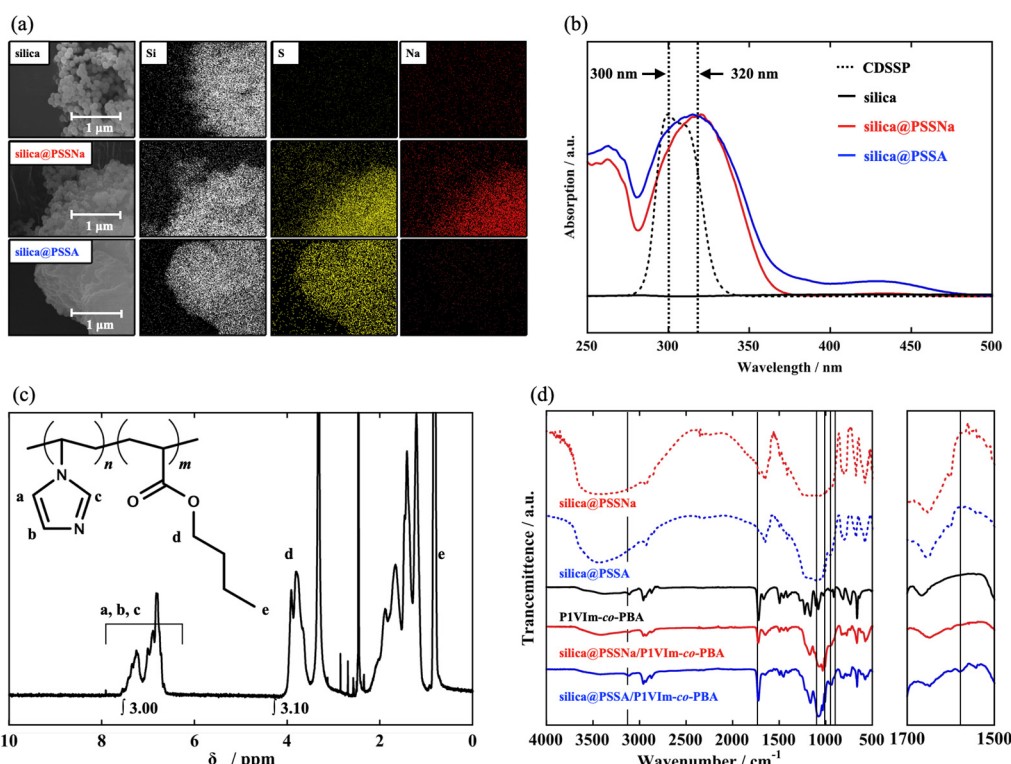

**Figure 1.** (**a**) SEM-EDX mappings of the pristine silica, silica@PSSNa, and silica@PSSA, (**b**) The Kubelka–Munk transformation of the reflectance curves of CDSSP, silica, silica@PSSNa, and silica@PSSA, and (**c**) [1]H NMR spectrum of P1VIm-*co*-PBA. The peaks a–e are corresponded to the hydrogen atoms of P1VIm and PBA. (**d**) FT-IR spectra of the silica@PSSNa, and silica@PSSA, P1VIm-*co*-PBA, silica@PSSNa/P1VIm-*co*-PBA, and silica@PSSA/P1VIm-*co*-PBA.

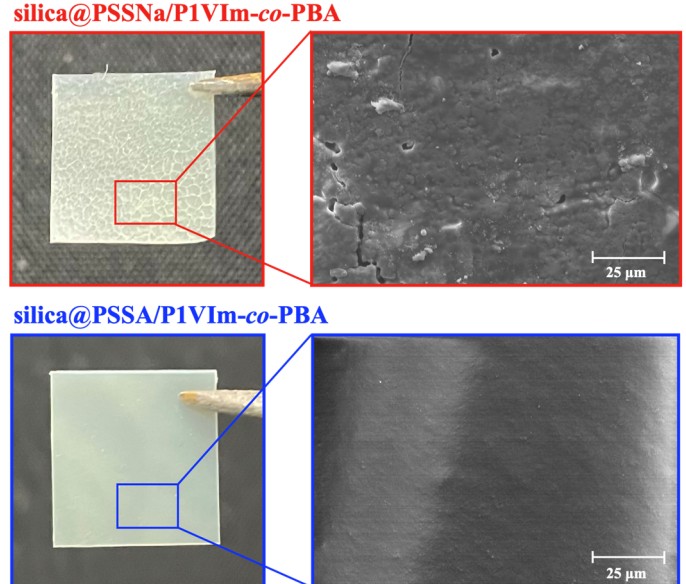

**Figure 2.** Photographs and SEM images of silica@PSSNa/P1VIm-*co*-PBA and silica@PSSA/P1VIm-*co*-PBA.

To evaluate the mechanical properties of each membrane, stress–strain curves were formulated, as shown in Figure 3a. Silica@PSSA/P1VIm-*co*-PBA achieved 17.0 MPa of strength, twice as high than that of silica@PSSNa/P1VIm-*co*-PBA, which is almost the same as that of the membranes in practical use [30,31]. The substantial increase can be attributed

to the robust acid-base interaction. The p$K_a$ values of SSA and 1VIm are 1.2–1.5 [32,33] and 6.0–6.5 [34], respectively, and the acid-base interaction attracts nanoparticles to the matrix, resulting in the suppression of voids which plays a crucial role; the direct connective network between nanoparticles and the matrix prohibits the formation of voids, which could act as potential breaking points [35]. Consequently, the stresses on the matrix are dispersed at the silica-matrix interface, resulting in a significant increase in mechanical strength. Therefore, the acid-base interaction significantly influences stress, proving beneficial for the fabrication of composite membranes with outstanding mechanical strength.

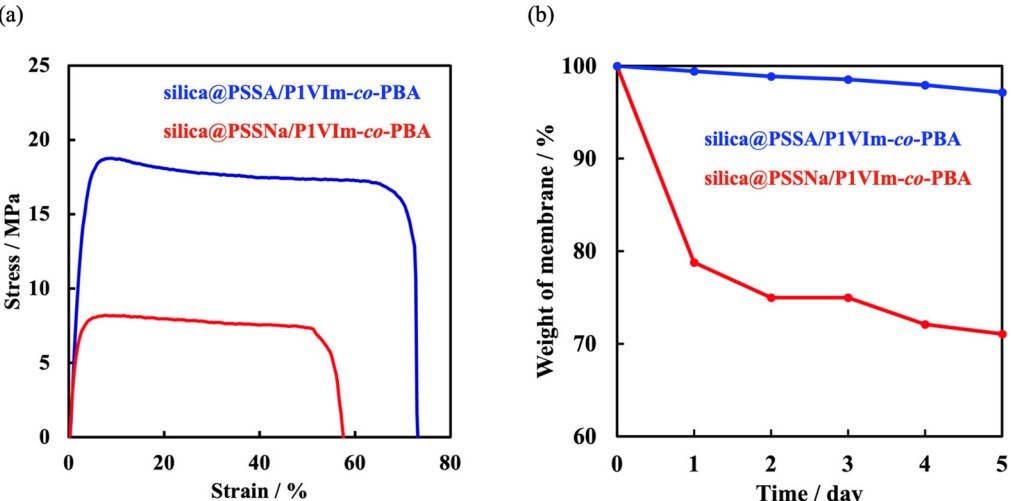

**Figure 3.** (**a**) Stress–strain and (**b**) retention stability of silica@PSSNa/P1VIm-*co*-PBA and silica@PSSA/P1VIm-*co*-PBA.

Generally, the isolated materials packed in the matrix leach from the membrane, affecting its long-term retention stability. Previous studies have reported a decrease in proton conductivity due to the excessive leaching of ionic liquid [36] and phosphoric acid [37] introduced into the matrix. Similarly, silica@PSSNa/P1VIm-*co*-PBA showed low-retention stability in the matrix, which does not form interactions between the nanoparticles and the matrix resulting in the leaching of the nanoparticles over time by soaking in water. The weight remaining rate of silica@PSSNa/P1VIm-*co*-PBA was 71.1 wt% after 5 days. On the other hand, silica@PSSA/P1VIm-*co*-PBA exhibited only a small amount of weight loss and the weight remaining after 5 days was 97.1 wt% (Figure 3b). Thus, the acid-base interaction suppressed the elution of nanoparticles, confirming their excellent long-term retention stability.

### 3.3. Proton Conductivities of Silica@PSSA, P1VIm-co-PBA and Silica@PSSA/P1VIm-co-PBA

Proton conductivities were calculated based on the obtained Cole–Cole plots shown in Figures S3–S5 The proton conductivities of silica@PSSA, P1VIm-*co*-PBA, and silica@PSSA/P1VIm-*co*-PBA at different temperatures (85% RH, 40–80 °C) are shown in Figure 4a and Table 2. The proton conductivity of silica@PSSA was measured in a pelletized state by compressing core–shell nanoparticles, and the proton conductivity was $1.54 \times 10^{-1}$ S/cm (80 °C, 85% RH). The results reveal a significant increase in the conductivity of silica@PSSA/P1VIm-co-PBA from 70 °C to 80 °C. This phenomenon was attributed to the glass transition temperature ($T_g$) of the composite membrane, which was observed at 75 °C by DSC measurement (Figure S6). For ion-conducting polymers, it has been reported that a lower $T_g$ is required for better performance [28,38]. The $T_g$ indicates that, as the temperature increases, ions in the membrane become more easily transported. Moreover, we observed a concurrent increase in water absorption within the same temperature range as the proton conductivity (Figure S7). This suggests that proton conduction was activated due to the heightened mobility of polymer chains and the enhanced movement of water.

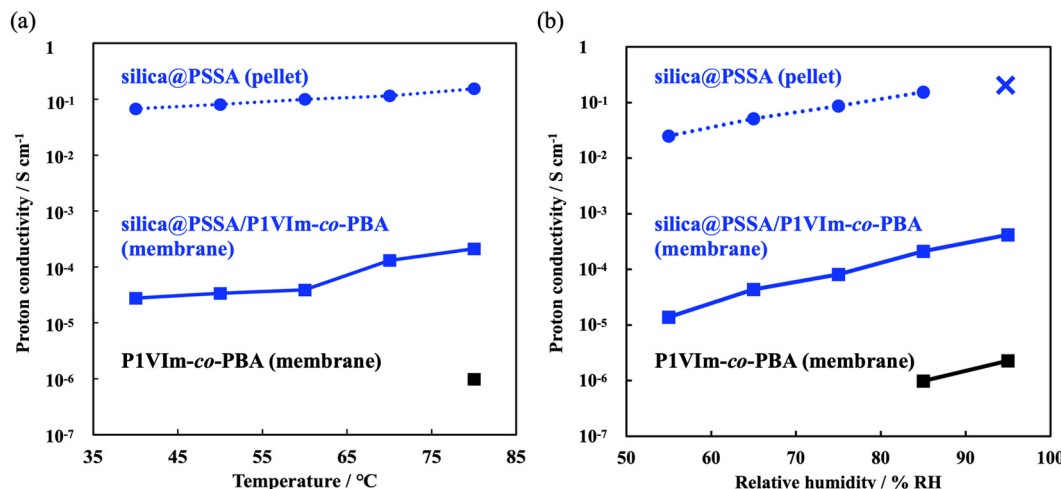

**Figure 4.** (**a**) At different temperature proton conductivity plots (85% RH) and (**b**) relative−humidity proton conductivity plots (80 °C) of the silica@PSSA (pellet), silica@PSSA/P1VIm-*co*-PBA (membrane), and P1VIm-*co*-PBA (membrane). "X" means not measurable.

**Table 2.** Summary of proton conductivities of silica@PSSA, silica@PSSA/P1VIm-*co*-PBA, and P1VIm-*co*-PBA.

| | 85% RH [S/cm] | | | | |
|---|---|---|---|---|---|
| | **40 °C** | **50 °C** | **60 °C** | **70 °C** | **80 °C** |
| silica@PSSA | $6.74 \times 10^{-2}$ | $8.05 \times 10^{-2}$ | $9.99 \times 10^{-2}$ | $1.15 \times 10^{-1}$ | $1.54 \times 10^{-1}$ |
| silica@PSSA/P1VIm-*co*-PBA | $2.77 \times 10^{-5}$ | $3.38 \times 10^{-5}$ | $3.88 \times 10^{-5}$ | $1.31 \times 10^{-4}$ | $2.11 \times 10^{-4}$ |
| P1VIm-*co*-PBA | - | - | - | - | $9.82 \times 10^{-7}$ |
| | 80 °C [S/cm] | | | | |
| | **55% RH** | **65% RH** | **75% RH** | **85% RH** | **95% RH** |
| silica@PSSA | $2.50 \times 10^{-2}$ | $5.10 \times 10^{-2}$ | $8.65 \times 10^{-2}$ | $1.54 \times 10^{-1}$ | - |
| silica@PSSA/P1VIm-*co*-PBA | $1.38 \times 10^{-5}$ | $4.37 \times 10^{-5}$ | $8.16 \times 10^{-5}$ | $2.11 \times 10^{-4}$ | $4.20 \times 10^{-4}$ |
| P1VIm-*co*-PBA | - | - | - | $9.82 \times 10^{-7}$ | $2.26 \times 10^{-6}$ |

In addition, relative-humidity-dependent proton conductivities are shown in Figure 4b (55–95% RH). Based on Figure 4a, the humidity-dependent measurement was performed at 80 °C, due to the fact that the temperature was above $T_g$ and high proton conductivity can be expected. However, the silica@PSSA pellet collapsed with increasing humidity due to moisture adsorption for the PSSA, and proton conductivity under 95% RH was not available (Figure S8). In contrast, silica@PSSA/P1VIm-*co*-PBA exhibited proton conductivity across all humidity ranges and achieved $4.20 \times 10^{-4}$ S/cm at 95% RH. Notably, this value was two orders of magnitude higher than that of bare P1VIm-*co*-PBA membrane. However, the proton conductivity of silica@PSSA/P1VIm-*co*-PBA was lower than that of silica@PSSA. This discrepancy is attributed to the sulfonic acid on the surface of silica@PSSA forming a bond with P1VIm, leading to a loss of proton-conducting performance and decreased water absorption [39]. Therefore, the sulfonic acid on the nanoparticle side acts as the proton conductor, and the contact points between the nanoparticles enhance proton conduction. The value of 40 wt% for silica@PSSA/P1VIm-*co*-PBA was optimized in a preliminary study. At 50 wt%, nanoparticle aggregations were observed and conductivity decreased. There is a possibility that aggregations inhibited proton conduction. Therefore, 40 wt% was determined to be the optimal ratio in this study (Table S1).

## 4. Conclusions

We have successfully fabricated a uniform composite membrane with high-density packing and high dispersibility of core–shell nanoparticles by applying acid-base interactions between the surface of nanoparticles and the matrix. Moreover, silica@PSSA/P1VIm-*co*-PBA demonstrated excellent mechanical strength, and the value matched well with that of the currently used membranes. Regarding proton conductivity, we achieved a substantial improvement by introducing nanoparticles to the P1VIm-*co*-PBA matrix, and the results in the proton conductivity were two orders of magnitude higher than that of the bare matrix. These outcomes were achieved by the contribution of the acid-base interaction of PSSA which was coated onto the surface of silica nanoparticles and basic P1VIm which was used as a matrix, by a simple mixing of core–shell nanoparticles and matrix. Therefore, these results indicate that the acid-base interaction at the surface of nanoparticles is one of the important factors for uniformly introducing nanoparticles for the membranes and providing the retention stability of nanoparticles for advancing applications in MMMs.

**Supplementary Materials:** The following supporting information can be downloaded at: https://www.mdpi.com/article/10.3390/technologies12020024/s1, Figure S1: Chemical structure of DIAION$^{TM}$; Figure S2: Cross-section SEM images (a) silica@PSSNa/P1VIm-*co*-PBA, and (b) silica@PSSA/P1VIm-*co*-PBA; Figure S3: Cole-Cole plots of silica@PSSA pellet at the (a) different temperature and (b) different relative humidity; Figure S4: Cole-Cole plots of silica@PSSA/P1VIm-*co*-PBA membrane at the (a) different temperature and (b) different relative humidity; Figure S5: Cole-Cole plots of P1VIm-*co*-PBA membrane at the different relative humidity; Figure S6: DSC curve of silica@PSSA/P1VIm-*co*-PBA; Figure S7: Water uptake of silica@PSSA/P1VIm-*co*-PBA; Figure S8: Photographs of silica@PSSA pellet at 55% RH and 95% RH; Table S1: SEM images and proton conductivities of different particle concentrations of silica@PSSA/P1VIm-*co*-PBA.

**Author Contributions:** Conceptualization, A.M.; Methodology, T.M., K.T., T.S., Y.M. and A.M.; Formal Analysis, T.M., K.T., T.S. and Y.M.; Writing—Original Draft Preparation, T.M.; Writing—Review and Editing, T.M., K.T. and A.M.; Supervision, A.M.; Funding Acquisition, A.M. All authors have read and agreed to the published version of the manuscript.

**Funding:** This research was funded by IZUMI Science and Technology Foundation, grant number 2022-J-010.

**Institutional Review Board Statement:** Not applicable.

**Informed Consent Statement:** Not applicable.

**Data Availability Statement:** Data are contained within the article and Supplementary Materials.

**Conflicts of Interest:** The authors declare no conflicts of interest.

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
