# Peer review of "High Affinity of Nanoparticles and Matrices Based on Acid-Base Interaction for Nanoparticle-Filled Membrane"

_technologies, doi:10.3390/technologies12020024_

Round 1

Reviewer 1 Report

Comments and Suggestions for Authors

By decorating the surfaces of silica nanoparticles with acidic polymers and integrating them into a basic matrix, PEMs were prepared for this manuscript. In addition to proton conductivity, the mechanical strength and retention stability of the membrane can be substantially enhanced. The manuscript is effectively structured and presented in its entirety. I propose that Technologies consider accepting this work with a minor revision. 

Despite the fact that a number of PEM parameters have been improved, their efficacy in realistic fuel cells still requires evaluation. The fuel cell efficacy of modified PEMs and commercial membranes should be compared by the authors. 

Reviewer 2 Report

Comments and Suggestions for Authors

In this submission, the authors report the fabrication of a composite membrane with core-shell nanoparticles incorporated via acid-base interactions. The proton conductivity of polymer matrix is significantly improved by these nanoparticles. In addition, the mechanical properties are improved. The topic is interesting and could attract wide readership from researchers working in the area of PEMs. Therefore, I recommend its publication after the following issues are addressed.   

1. What’s the chemical composition of DIAION? What’s the mechanism for the cation exchange using DIAION?

2. The authors are advised to provide the cross-section SEM image or TEM image of the composite membrane to show the nanoparticle distribution in it.

3. There is no proton conductivity value for silica@PSSNa/P1VIm-co-PBA.

4. What’s the evidence for the acid-base interaction? Is the enhancement of membrane performance caused by other interactions?

5. The authors are recommended to cite relevant literatures such as RSC Adv. 2014, 4, 46265 and Nanoscale 2021, 13, 18332.

Comments on the Quality of English Language

The language is OK. 

Reviewer 3 Report

Comments and Suggestions for Authors

This paper prepared PEMs by coating the surfaces of silica nanoparticles with acidic polymers and integrating them into a basic matrix. Due to the acid-base interaction between the nanoparticles and the matrix, leading to composite membranes with a high dispersion and densely packed nano particles. This fabrication technique significantly improved mechanical strength and retention stability. The paper is interesting and meaningful. This paper can be accepted after a minor revision.

1 The performance comparison of this membrane with Nafion can be given in Table 2.

2 The silica@PSSA (0.200 g) was dispersed and P1VIm-co-PBA (0.300 g) was used to cast the membrane. Is the ratio of 2/3 the optimized ratio? Maybe more silica@PSSA can be used to enhance the proton conductity.

Comments on the Quality of English Language

Minor editing of English language required

Reviewer 4 Report

Comments and Suggestions for Authors

1. Scheme 1 should be improved to show the chemical structure of the molecules on the nanoparticle's surface.  As well as the possible interaction with the membrane copolymer.

2. The clear evidence of the proton transfer from the functionalized silica nanoparticles to the P1VIm-co-PBA is scarce.

a) Assigning a very weak peak in the FTIR (1580cm-1) spectra to that interaction is insufficient. That peak is much more intense in Ref 28. How do the authors explain the very weak intensity of that peak?

b) Furthermore, the authors assign the peak close to 1580 to ImH+, but that signal is 50 cm-1 shifted from the value reported in ref 28. This significant shift in the position of the peak must be explained.

c) I strongly recommend that the authors do a deeper analysis of the FTIR spectrum to clarify the interaction between the functionalized nanoparticles and the membrane

3. The statement in line 245 must be proved with additional experiments. In another way, the sentence should be rewritten, suggesting what could be the cause of that discrepancy.

Comments on the Quality of English Language

1. The sentence in line 151 should be rewritten.

2. The sentence on line 238 must be rewritten. The verb is missing.

3. Was the proton conductivity under 95% RH not available, or was it not possible to measure it?

Round 2

Reviewer 4 Report

Comments and Suggestions for Authors

The paper can be accepted without any further changes.

Comments on the Quality of English Language

The manuscript has an acceptable quality of English